# Leveraging Learning Rate Gradients for Automatic Learning Rate Selection

## Abstract

Selecting an optimal learning rate (LR) is crucial for training deep neural networks, significantly affecting both convergence speed and final model performance. Determining this optimal LR typically involves two key challenges: choosing an appropriate initial LR and selecting an LR scheduler for adjusting the LR during training. This paper focuses on the former challenge—selecting the initial LR. Traditionally, this task relies on manual tuning or heuristic methods, often involving extensive trial-and-error or computationally expensive search strategies like grid search or random search. We propose an algorithm, Automatic Learning Rate Selection (ALRS), to find the initial LR without the need for manual intervention. ALRS leverages the gradient of the LR itself — a less explored approach in the field. ALRS is a computationally lightweight pre-training process that automatically selects the initial LR by iterative refinements using the LR gradient, specifically analyzing its sign information, combined with suitable search algorithms. This approach efficiently converges to the optimal LR in a stable and robust manner across various optimizers and network architectures.

We evaluate our technique on standard deep learning benchmarks, including MNIST with a CNN and CIFAR-10 and CIFAR-100 with ResNet-18, using both SGD and Adam optimizers. Our experiments demonstrate that the automatically determined LRs achieve performance comparable to manually tuned LRs and state-of-the-art results.

## 1 Introduction

In recent years, deep learning has revolutionized numerous fields, from image recognition and natural language processing to game playing and scientific discovery. At the heart of these advancements lies the optimization process, where the objective is to minimize a loss function through iterative updates of the model's parameters. This optimization process is driven by optimizers, such as Stochastic Gradient Descent (SGD) and Adaptive Moment Estimation (ADAM) (Ruder, 2016), which play a crucial role in determining the direction and magnitude of parameter updates.

One of the most critical hyperparameters in this optimization process is the learning rate (LR) (Bengio, 2012). The LR controls the step size taken in the direction of the gradient, significantly influencing the convergence speed and stability of the training process. A LR that is too high can cause the model to oscillate or even diverge, while a LR that is too low can lead to slow convergence and suboptimal solutions (Yang & Shami, 2020). Therefore, selecting an appropriate initial LR is essential for efficient and effective training of neural networks. In addition to choosing an initial LR, one often needs to choose a scheduler that adapts the LR during the training process. Choosing a scheduler can be thought of as a separate problem; the sole focus of this paper is the problem of choosing the initial LR.

### 1.1 Challenges in LR Selection

Traditionally, the selection of the LR has essentially been a manual process, often involving extensive trial and error. Researchers and practitioners typically perform grid search or random search over a predefined set of possible values, followed by fine-tuning based on empirical observations (Bischl et al., 2023; Yang & Shami, 2020). This process is time-consuming and computationally expensive. Additionally, the optimal LR can vary significantly across different datasets, architectures,

and even different stages of training, further complicating the selection process (Nar & Sastry, 2018; Jastrzębski et al., 2017).

Techniques such as LR schedulers (e.g., step decay, exponential decay) adjust the LR during training based on predefined rules or gradients of the parameters (Johnson et al., 2024). While these methods offer improvements over fixed LRs, they still depend on initial LR settings, thereby making the choice of initial LR crucial. The dependence of this initial LR on the model, dataset and optimizer makes the choice of initial LR challenging. The challenge that we aim to address in this paper is finding the initial LR in a computationally lightweight manner.

### 1.2 OUR CONTRIBUTIONS

In this paper, we propose an algorithm called Automatic LR Selection (ALRS) (see Section 3) to automatically determine the optimal initial LR. The main ingredient of our approach is the gradient of the LR itself. The use of LR gradient appears in (Almeida, 1998; Baydin et al., 2017). However, we do not directly apply gradient descent to the LR, thus deviating from prior hypergradient descent methods.

Specifically, our contributions include:

1. **ALRS algorithm**: We calculate the gradient of the loss function with respect to the LR. Rather than following the hypergradient descent method, we propose an algorithm that produces increase and decrease signals depending essentially on the signs of these gradients. The problem of adjusting the LR based on these signs calls for separate search algorithms. We propose an implementation of one such search algorithm (see Section 3). ALRS is a pre-training process with computational costs typically equal to that of one training epoch.

2. **Empirical Validation**: We validate our approach through experiments on three benchmark datasets: MNIST with a Convolutional Neural Network (CNN), CIFAR-10 with ResNet-18, and CIFAR-100 with ResNet-18. Our method demonstrates robust performance across different combinations of optimizers and models, achieving results at par with state-of-the-art performance.

3. **Study of the dependence of ALRS on initial seeds**: The ALRS depends on two seeds: a random seed and a starting LR seed from which the optimal LR is derived. We study the dependence of ALRS on these seeds and show that it is within acceptable limits.

By addressing the challenge of automatic initial LR selection, this work aims to contribute to advancements in finding hyperparameter-less training algorithms in the field of deep learning.

### 1.3 STRUCTURE OF THE PAPER

The remainder of this paper is organized as follows: Section 2 provides a review of related work in hyperparameter optimization and LR adjustment methods. Section 3 details the methodology of ALRS, including search algorithms. In Section 4, we present the experimental setup and results, showcasing the effectiveness of ALRS. Section 5 studies the dependence of ALRS on initial seeds. Section 6 offers a discussion on the limitations of our approach and directions for future work.

## 2 RELATED WORK

### 2.1 HYPERPARAMETER OPTIMIZATION

Hyperparameter Optimization (HPO) is a crucial aspect of machine learning that involves selecting the best set of hyperparameters for a given model. Hyperparameters are parameters whose values are set before the learning process begins, apart from LR some other hyperpatameters such as batch size, and the number of layers in a neural network are also optimized via HPO.

Several methods exist for HPO, including Grid Search, a brute-force approach that exhaustively searches through a predefined set of hyperparameters. In this approach, an exhaustive search through all possible combinations can be very time-consuming and computationally expensive. Random Search randomly samples hyperparameters from a specified distribution (Bergstra & Bengio, 2012).

This approach may require a large number of iterations to find optimal hyperparameters, as it does not leverage any information from previous evaluations. Also, it does not systematically explore the hyperparameter space, potentially missing optimal regions. Bayesian Optimization (Wu et al., 2019; Victoria & Maragatham, 2021) uses probabilistic models to predict the performance of hyperparameters and iteratively refines the search space. This probabilistic model of the objective function can be complex to implement and tune. Hypergradient descent (Baydin et al., 2017; Almeida, 1998; Donini et al., 2019; Franceschi et al., 2017), which deals with continuous hyperparameters like LR, is closest to our approach; we discuss the similarities and differences in the following subsection.

## 2.2 Hypergradient Descent Vs ALRS

Hypergradient descent (HD) uses the gradient method to fine tune the LR. The similarity between ALRS and HD is that they both use the gradient of the LR. However, the following are the key differences:

1. Unlike HD, ALRS does not use the gradient descent method to fine-tune the LR. Instead, a separate algorithm that essentially uses signs of this gradient across batches is used.

2. Unlike HD, ALRS is a pre-training algorithm to find the optimal initial LR, after which its role is over. Hence, we expect ALRS to be free from overfitting issues seen in the case of HD (Jin et al., 2021).

3. HD appears to be sensitive to newly introduced parameters in HD viz. Hypergradient LR and also the initial starting LR. ALRS is not sensitive to newly introduced parameters like LR seed, see Section 5.

## 3 ALRS

The ALRS algorithm consists of two separate sub-algorithms. The goal of the first sub-algorithm is to point the direction in which the current LR should be adjusted to approach an optimal value, i.e., to come up with increase and decrease signals. The signs of the LR gradient, as well as the total magnitude across a set number of batches, are used to come up with these signals. For ease of illustration, we think of the second sub-algorithm as being implemented by an instance of a ParameterSearch class. This class is initialized with a starting or a seed value of a parameter (in our case, LR) and responds to the increase and decrease signals via increase and decrease methods, which adjust the parameter value by magnitude depending on the class implementation. The goal of this ParameterSearch class is to converge to an optimal parameter value. So, the class also provides a method (called done) that detects convergence within a specified tolerance.

We now explain the ALRS algorithm, which takes an instance of the parameter class as an input. In the subsequent subsection, we outline a specific implementation of the ParameterSearch class, which we call SBF search.

### 3.1 ALRS algorithm

While the precise algorithm is given in Algorithm 1, we briefly describe the main ideas used in ALRS. It is important to note that ALRS being a pre-training process the weights of the model are not modified throughout the algorithm. We use an instance of an optimizer denoted by OPT (like Adam or SGD). For each training batch, we run the OPT.step() method which temporarily modifies the weights of the model. However, we store the old weights in a variable and not only restore the original weights but also save the difference in a dictionary, which is denoted by ADJUSTMENTS. This dictionary is initialized to zero in the beginning. In each training batch, instead of using the actual weights of the model we use WEIGHT $+ \gamma \cdot$ ADJUSTMENT, where $\gamma$ is a variable initialized to 1. This allows us to calculate the gradient of the loss function w.r.t. $\gamma$. Since the ADJUSTMENTS are proportional to the LR for optimizers like Adam or SGD, this gradient is interpreted as the gradient of the loss function w.r.t. LR. An increase signal is generated if $90\%$ of the LR-gradients for a specified number of steps are negative. In our case, this specified number of steps is total training batches divided by 50; the algorithm appears robust towards small changes in this value. The reason for looking at multiple batches before generating a signal is that a single training batch may not be sufficiently representative of the data. If an increase signal is not generated, then by default

---

**Algorithm 1:** ALRS

---

**Input :** (1) MODEL
      (2) OPT: an instance of an optimizer like Adam or SGD
      (3) LR_SEARCH: an instance of ParameterSearch class (see section 3.2)
      (4) num_batches per epoch
**Initialization:**
    1. LR_SEARCH.value = lr = 1

    2. $\gamma = 1$

    3. num_steps = int(num_batches / 50)

    4. LR_GRADS = []

    5. dict(ADJUSTMENTS) = 0· MODEL.named_parameters

**while** *not* LR_SEARCH.*done()* **do**
    dict(OLD_PARAMS) ← MODEL.*named_parameters*
    **for** *name, p in* MODEL.*named_parameters* **do**
      | p ← p + $\gamma$· ADJUSTMENTS[name]
    **end**
    OPT.step()
    LR_GRADS.append($\nabla_\gamma$ (loss ))
    **for** *name, p in* MODEL.*named_parameters* **do**
      ADJUSTMENT[name] ← p - OLD_PARAMS[name]
      p ← OLD_PARAMS[name]
    **end**
    **if** *len(*LR_GRADS*) == num_steps* **then**
      num_neg_grads = number of negative elements in lr_grads
      total_grad = sum(lr_grads)
      **if** *num_neg_grads >= 0.9 · num_steps AND total_grad < 0* **then**
        | LR_SEARCH.increase()
      **end**
      **else**
        | LR_SEARCH.decrease()
      **end**
      LR_GRADS = []
      lr ← LR_SEARCH.value
    **end**
**end**
lr ← lr / 10

---

a decrease is generated. Finally, these signals are fed to the ParameterSearch class like SBF search
(see Section 3.2). Empirical evidence shows that the actual value obtained by the ParameterSearch
class when divided by 10 is close to the optimal initial LR that we seek. Section 6.1 contains more
discussion on the choice of this number $1/10$.

## 3.2 SBF SEARCH

ALRS requires an implementation of a ParameterSearch class (see Section 3). Recall that a Param-
eterSearch class is initialized with a starting seed value of the parameter and responds to increase
and decrease signals with the goal of converging to an optimal value of the parameter. The imple-
mentations of interest in this paper utilize the concept of MOMENTUM, which is calculated as an
exponential moving average of +1 (for increase signals) and -1 (for decrease signals). The role of
MOMENTUM is to quantify the oscillatory behaviour of the signals.

SBF Search (SBFS) is a particular implementation of the ParameterSearch class that is used for run-
ning experiments in this paper. Other implementations are possible (see Section 6). SBFS combines
two separate implementations of ParameterSearch class, starting with Soft Binary Search (SBS) and
switching later to Fractional Search (FS) for better convergence when the parameter value is within
a certain range.

SBS is a modified binary search algorithm that uses soft limits instead of hard boundaries. These
limits are called soft limits because, after each increase signal, the upper limit is increased depending
on the size of the MOMENTUM.

FS is a search algorithm that adjusts the value by a fraction. The fraction depends on the MOMEN-TUM, increasing it for consecutive moves in the same direction and decreasing it for oscillatory moves.

SBFS combines these two algorithms in the following manner:

1. Initialization: The algorithm initializes SBS with the initial parameter value.

2. SBS Phase: SBS is used to narrow down the search space. The soft limits of SBS are iteratively adjusted on each increase or decrease signal using a scaling factor between 1.05 and 1.5 depending on magnitude of MOMENTUM.

3. Switch to FS: When the difference between the upper and lower limit contracts to less than 5% of the parameter value or after 100 signals, the algorithm switches to FS.

4. FS Phase: FS is used to refine the search further, adjusting the parameter value by a small fraction, which is between 1% and 5% depending on the magnitude of the MOMENTUM.

5. Termination: We run FS for 20 iterations.

## 4 EXPERIMENTS

### 4.1 EXPERIMENTAL SETUP

We evaluate our approach using standard image classification benchmarks widely used in the literature: MNIST (with a custom CNN), CIFAR-10, and CIFAR-100 (Vinod & Geoffrey) (both using ResNet-18 He et al. (2015)). The models are trained for 10, 50, and 50 epochs, respectively, using the LRs obtained by ALRS. We employ two optimizers in our experiments: SGD with a momentum of 0.9 and Adam with default settings. We compare the results with those obtained using commonly used LRs from the literature (Zhang et al., 2019). It is noteworthy that in standard CIFAR-10 and CIFAR-100 training reported in the literature, weight decay is typically used; however, we do not use weight decay in our experiments.

To focus on the optimality of the initial LR selection and separate it from the effects of LR schedulers, we avoid full training for CIFAR-10 and CIFAR-100. This allows us to isolate the impact of the choice of initial LR.

Experiments are conducted using three different random seeds, and the training batch size is set to 128 for all models using default datasets. The training loss at the end of chosen number of epochs is displayed in Tables 1, 2, 3.

By default, ALRS starts with an initial LR seed of 1 and then fine-tunes it. To study the impact of different initial LR seeds and random seeds, we conducted experiments using ten different initial LR seeds in the range [0.001, 10], each evaluated across three different random seeds (see Table 4).

### 4.2 RESULTS

We compare the result of the training using the LR predicted by ALRS with standard LRs. By standard LR we mean LRs, which have been used in literature to achieve results close to benchmarks (see Zhang et al. (2019)). The experimental results demonstrate that the LR predicted by ALRS achieves performance comparable to the standard LRs. For each model/optimizer combination, we present graphs displaying the following lines on log scale:

1. Training loss using the standard LR (mean over 3 random seeds).

2. Training loss using the LR predicted by ALRS (mean over 3 random seeds).

3. Training losses using standard LR scaled by factors of 10 and 0.1 (mean over 3 random seeds).

## 5 DEPENDENCE OF ALRS ON SEEDS

We investigated how different choice of random seeds and initial LR seeds affect the LR predicted by ALRS. Although we recommend starting ALRS with an initial LR seed of 1, we ran the algorithm

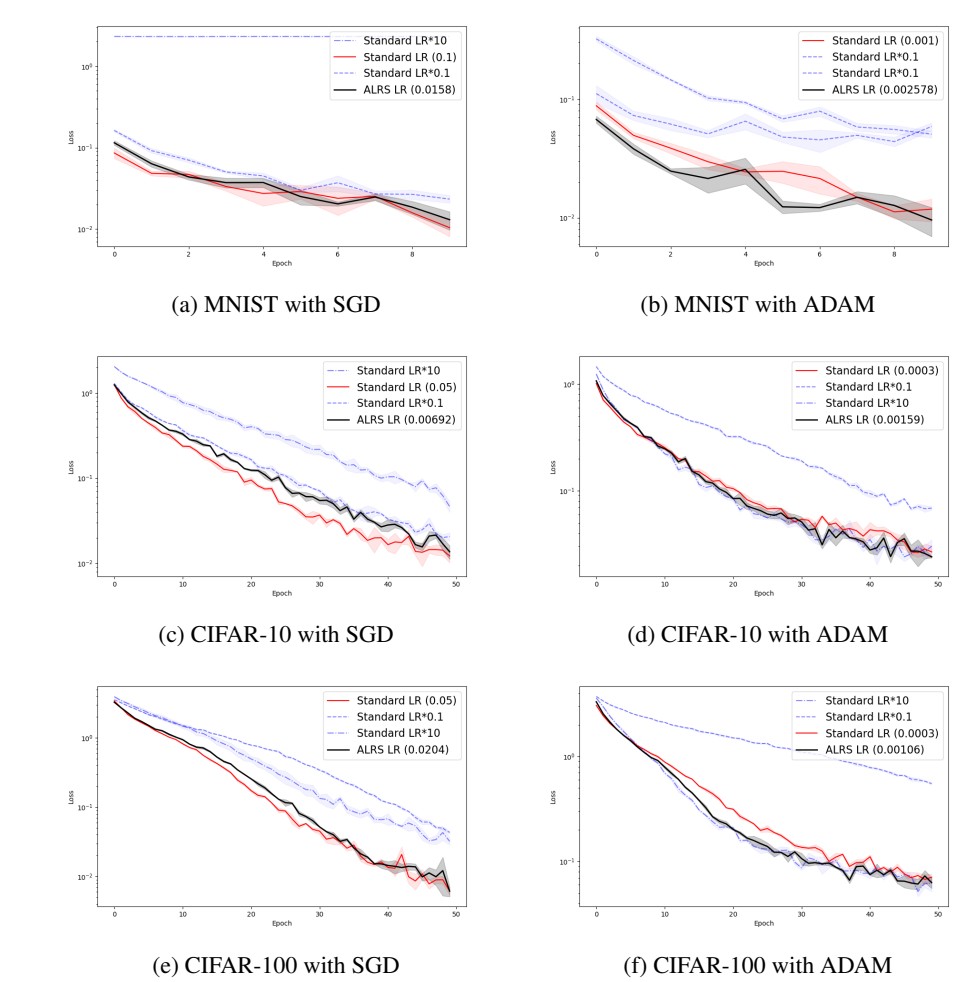

(a) MNIST with SGD

(b) MNIST with ADAM

(c) CIFAR-10 with SGD

(d) CIFAR-10 with ADAM

(e) CIFAR-100 with SGD

(f) CIFAR-100 with ADAM

Figure 1: Comparison of training loss for n epochs between LR found by ALRS, standard LR, standard LR*0.1, and standard LR*10 for MNIST (n=10), Cifar-10 (n=50), Cifar-100 (n=50) with ADAM, SGD optmizers

Table 1: MNIST:Training loss (10 Epochs)

| Optimizer | LR | Loss |
|---|---|---|
| SGD | Std | $0.0104 \pm 0.0023$ |
| | ALRS | $0.0131 \pm 0.0033$ |
| ADAM | Std | $0.0118 \pm 0.0025$ |
| | ALRS | $0.0096 \pm 0.0026$ |

using 10 different initial LR seeds in the range [5e-5, 10], each evaluated across 3 different random seeds. The variations in the LR calculated by ALRS resulting from these choices are presented in Table 4. The results demonstrate the robustness of the algorithm for different seed choices. Additionally, the effect of variation in the LR due to the random seed can be observed as the standard error plotted in Figure 1.

Table 2: Cifar-10:Training loss (50 Epochs)

| Optimizer | LR | Loss |
|---|---|---|
| SGD | Std | $0.0122 \pm 0.0020$ |
| | ALRS | $0.0137 \pm 0.0025$ |
| ADAM | Std | $0.0268 \pm 0.0036$ |
| | ALRS | $0.0241 \pm 0.0004$ |

Table 3: Cifar-100:Training loss (50 Epochs)

| Optimizer | LR | Loss |
|---|---|---|
| SGD | Std | $0.0062 \pm 0.0008$ |
| | ALRS | $0.0061 \pm 0.0010$ |
| ADAM | Std | $0.0707 \pm 0.0055$ |
| | ALRS | $0.0629 \pm 0.0075$ |

Table 4: Learning Rates Predicted by ALRS Across Different Initial LR Seeds and Random Seeds

| Model | Optmizer | mean LR | Std error | % variation |
|---|---|---|---|---|
| MNIST | ADAM | 0.001698 | 3.8e-4 | 22.37% |
| | SGD | 0.016427 | 2.6e-3 | 15.90% |
| Cifar-10 | ADAM | 0.001874 | 2.2e-4 | 11.96% |
| | SGD | 0.006928 | 1.5e-4 | 2.09% |
| Cifar-100 | ADAM | 0.001123 | 6.5e-5 | 5.83% |
| | SGD | 0.020934 | 4.5e-4 | 2.16% |

# 6 CONCLUSION, LIMITATIONS AND FUTURE DIRECTIONS

## 6.1 LIMITATIONS AND FUTURE DIRECTIONS

Despite the effectiveness of ALRS, certain limitations of this work highlight important areas for future research:

1. **Alternative ParameterSearch Methods**: Implementing a ParameterSearch class based on reinforcement learning as an alternative to the SBFS (see 3) could potentially yield better performance. This approach warrants further investigation to assess its viability and effectiveness.

2. **The choice of** $1/10$: ALRS uses an LR that is one-tenth of the value obtained from the ParameterSearch class. While the optimal LR found directly from the ParameterSearch class can be considered ideal for training the next few batches, determining the optimal LR after every few batches is computationally expensive. Therefore, using a fraction of the found LR becomes necessary. In this study, a fraction of 1/10 was used, which proved effective across tested configurations. However, a larger fraction could be employed at the expense of more frequent scheduling. The choice of this fraction is closely related to the problem of LR scheduling.

3. **LR Scheduling**: The issue of scheduling the LR throughout the entire training process remains unaddressed. Applying similar ideas to automate LR adjustments over time could lead to a fully automated training process, which is an important direction for future work.

4. **Testing on larger models** : ALRS remains untested on language models, including large language models (LLMs). Evaluating its performance on such models is essential to understand its scalability and applicability in more complex scenarios.

## 6.2 CONCLUSION

Leveraging the LR gradient offers a promising solution for determining the optimal initial LR when training deep neural networks. Our experiments demonstrate that ALRS is effective in automating this selection process. We note that, the LRs predicted by ALRS do not necessarily align with standard values; in some cases, they are significantly larger or smaller. Despite these differences, the training loss achieved using ALRS is comparable to that obtained with standard LRs, highlighting the method's success. This confirms that automating the initial LR selection through ALRS is both feasible and beneficial, enhancing training efficiency and effectiveness. An important direction for future work is to develop a suitable LR scheduling algorithm that can work in tandem with ALRS, further improving and fully automating the training process.

AUTHOR CONTRIBUTIONS

Both authors have contributed equally to the paper. We follow the mathematical convention of ordering the authors alphabetically.

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
