# OpenReview forum: "LEVERAGING LEARNING RATE GRADIENTS FOR AUTOMATIC LEARNING RATE SELECTION"
_ICLR.cc/2025/Conference — ICLR 2025 Conference Withdrawn Submission_

### Official Review · Reviewer_h5fn · 2024-10-28

**Soundness:** 1
**Presentation:** 1
**Contribution:** 1
**Rating:** 3
**Confidence:** 4

**Summary:**

This paper proposes a method that uses hypergradients as a surrogate signal to search for the initial learning rate.
ALRS uses hypergradient signs to guide initial learning rate search through a two-stage process of soft binary and fractional search.

**Strengths:**

-	This paper leverages hypergradients as surrogate signals to determine initial learning rates through a two-stage search process combining soft binary and fractional search.

**Weaknesses:**

-	The paper lacks comparisons with other hypergradient-based methods, despite using hypergradients as a surrogate signal. Adding such comparisons could clarify ALRS’s strengths and limitations relative to existing methods.
-	This paper would benefit from more comprehensive experiments demonstrating budget efficiency, scalability across different architectures, and performance gains across a wider range of benchmarks. Including models like GPT-2 or RNNs in language tasks would strengthen the evaluation. Additionally, clarifying budget efficiency by comparing ALRS’s runtime and memory usage against other methods would provide a more practical view of its efficiency.
-	The paper only focuses on initial learning rate selection while HPO methods usually handle multiple interacting hyperparameters that affect each other's optimal values. Consider either discussing this limitation or suggesting how ALRS could be extended for broader HPO tasks

**Questions:**

Please refer to the weakness section.

---

### Official Review · Reviewer_8tGi · 2024-10-28

**Soundness:** 2
**Presentation:** 1
**Contribution:** 2
**Rating:** 3
**Confidence:** 3

**Summary:**

The paper proposes the Automatic Learning Rate Selection (ALRS) algorithm to determine an optimal initial learning rate without manual tuning. The approach utilizes the signs of the gradient of the loss function with respect to the learning rate for this selection process. It is much cheaper than trial-and-error approaches, which require multiple training steps. The ALRS algortihm is evaluated for different data sets, models, and optimizers.

**Strengths:**

- The need for an algorithm to determine the learning rate automatically is omnipresent.
- The algorithm itself appears to be working.

**Weaknesses:**

Unfortunately, the paper's text needs major rework:
- Please cite the original publication of Adam in the introduction when Adam is first mentioned.
- The text contains many repetitions of already mentioned statements. This can be helpful to recapitulate important statements, but my feeling is that the text could be shortened without loosing information/clarity.
- Section 3.1 is formulated in a very technical way. I am aware that it describes an algorithm, but the main ideas should not be described by sentences like: "For each training batch, we run the OPT.step() method..." or "However, we store the old weights in a variable and not only restore the original weights but also save the difference in a dictionary, which is denoted by ADJUSTMENTS." Please try to describe the ideas and do not transform your code into a text. The reader of the paper is probably more interested in how this algorithm works than how you implemented this algorithm. If one is interested in the implementation, one can look into the code. Moreover, I think that first describing the algorithm and afterwards naming the default values of the ALRS hyperparameters is easier to follow.
- Please add the citation of the MNIST data set when MNIST is first mentioned.
- The discussion of the results is missing. The results are only presented in figures/tables without a discussion what you conclude from them.

**Questions:**

- How did you determine the hyperparameter values of ALRS?
- The ALRS algorithm requires to set a lot of hyperparameters itself to find the learning rate. Please comment on how general are your recommendations?
- Why did you not use a weight decay?

---

### Official Review · Reviewer_XEdo · 2024-10-30

**Soundness:** 2
**Presentation:** 3
**Contribution:** 1
**Rating:** 3
**Confidence:** 5

**Summary:**

The paper presented a new algorithm to select a proper initial learning rate. It uses the gradient of the loss w.r.t the learning rate itself to select the initial learning rate. The algorithm is tested on Mnist, Cifar10, Cifar100 dataset with Resnet18, ResNet100, and showed comparable results with manually selected learning rate.

**Strengths:**

The author studied an interesting question, the learning rate selection is an interesting topic in deep learning.

The overall methodology is intuitive to understand and author did a decent job describing it in the paper.

**Weaknesses:**

In short, I think the paper presents an overly complicated approach to solve the learning rate selection problem. As a result, it significantly reduced the practical value of the paper, to even defeats the purpose of the value add.

The proposed algorithm requires storing another sets of parameter for every weight, which significantly increased the memory requirement. This will also introduce non-negligible speed penalty too.   Furthermore, calculating the gradient of loss w.r.t the learning rate itself requires persistent gradient tracking, which further adds burden to the speed and memory usage.   The paper only describes the convergence speed in terms of epoch, but have no mention of the run time speed and memory usage.   If the proposed algorithm is significantly slower than even manually selecting the learning rate, then the goal of the paper is heavily undermined.

**Questions:**

1. the division of optimal LR by 10 is a bit unsettling to see, despite what author had explained in 6.1.  Choosing an LR then decrease it by an entire order of magnitude is almost saying the LR found isn't really optimal.   Can author provide any theoretical backing of such action?


2. can the author elaborate more on how LR changes after LR.increase() and LR.decrease()?

---

### Official Review · Reviewer_AYaa · 2024-11-04

**Soundness:** 2
**Presentation:** 3
**Contribution:** 2
**Rating:** 3
**Confidence:** 4

**Summary:**

This paper addresses the challenge of selecting the initial LR, traditionally managed through manual tuning or costly search methods. It introduces Automatic Learning Rate Selection (ALRS), an algorithm that determines the initial LR without manual intervention. ALRS uses the gradient of the LR itself, performing lightweight pre-training to iteratively refine the LR based on its sign and targeted search techniques.

**Strengths:**

1. The paper is well-written, with a clear and easy-to-follow introduction and related work sections.

2. The issue of Auto Learning Rate is clearly defined, and the paper's contributions are explicitly highlighted.

3. The overall concept of the paper is engaging, with a clear logic and motivation. Moreover, the algorithm's use of gradient signs to generate increase and decrease signals adds an interesting aspect.

**Weaknesses:**

1. The description in Algorithm 1 is generally acceptable, but much of it reads more like pseudocode, particularly elements like *LR_SEARCH.done()*. It’s highly recommended to use mathematical notation for a more universally applicable approach.

2. The paper is validated only on the CIFAR-10 and CIFAR-100 dataset. Learning rate scheduling is a complex problem, and using ResNet-18 on CIFAR-10/100 is a relatively simple experiment, making it challenging to convincingly demonstrate the algorithm’s effectiveness.

3. The comparisons are insufficient. The proposed method is only evaluated against standard approaches. Given the related work discussed, why not compare with other state-of-the-art methods?

4. One reason Hyperparameter Optimization performs well is that it leverages gradients with respect to the hyperparameters. Since the proposed method introduces a different approach, is there any evidence supporting its effectiveness?

**Questions:**

1. It is well-known that automatic learning rate selection can encounter issues with short-horizon search [1]. How does this algorithm address or mitigate that challenge?

2. Minor format issue: there are two paragraphs in the abstract.

[1] Wu, Y., Ren, M., Liao, R., & Grosse, R. (2018). Understanding short-horizon bias in stochastic meta-optimization. arXiv preprint arXiv:1803.02021.

---

### Note · Authors · 2024-11-25

**Comment:**

We thank the reviewers for their constructive feedback. We are withdrawing the paper since we need a substantial amount of time to revise the papers as per these suggestions.

**Withdrawal Confirmation:**

I have read and agree with the venue's withdrawal policy on behalf of myself and my co-authors.